# Antiviral Functions of Type I and Type III Interferons in the Olfactory Epithelium

**DOI:** 10.3390/biom13121762

**Published:** 2023-12-08

**Authors:** Ahmad Zedan, Ashley D. Winters, Wei Yu, Shuangyan Wang, Ying Ren, Ashley Takeshita, Qizhi Gong

**Affiliations:** 1Department of Cell Biology and Human Anatomy, School of Medicine, University of California, Davis, CA 95616, USA; azedan@ucdavis.edu (A.Z.); adwinters@ucdavis.edu (A.D.W.); aatakeshita@ucdavis.edu (A.T.); 2Department of Physiology, Xi’an Medical University, Xi’an 710021, China; yuwei@xiyi.edu.cn; 3Department of Human Anatomy, Histology and Embryology, School of Basic Medicine, Qingdao University, Qingdao 266071, China; wangshuangyan@qdu.edu.cn; 4Department of Stomatology, Zhongshan Hospital, Fudan University, Shanghai 200032, China; ren.ying@zs-hospital.sh.cn

**Keywords:** innate immunity, antiviral response, olfaction

## Abstract

The olfactory neuroepithelium (OE) is one of the few neuronal tissues where environmental pathogens can gain direct access. Despite this vulnerable arrangement, little is known about the protective mechanisms in the OE to prevent viral infection and its antiviral responses. We systematically investigated acute responses in the olfactory mucosa upon exposure to vesicular stomatitis virus (VSV) via RNA-seq. VSVs were nasally inoculated into C57BL/6 mice. Olfactory mucosae were dissected for gene expression analysis at different time points after viral inoculation. Interferon functions were determined by comparing the viral load in interferon receptor knockout (*Ifnar1*^−/−^ and *Ifnlr1*^−/−^) with wildtype OE. Antiviral responses were observed as early as 24 h after viral exposure in the olfactory mucosa. The rapidly upregulated transcripts observed included specific type I as well as type III interferons (*Ifn*) and interferon-stimulated genes. Genetic analyses demonstrated that both type I and type III IFN signaling are required for the suppression of viral replication in the olfactory mucosa. Exogenous IFN application effectively blocks viral replication in the OE. These findings reveal that the OE possesses an innate ability to suppress viral infection. Type I and type III IFNs have prominent roles in OE antiviral functions.

## 1. Introduction

The olfactory neuroepithelium (OE) is one of the few neural tissues that directly interact with the environment. The olfactory mucosa (OM), located posteriorly in the nasal cavity, consists of the OE and underlining lamina propria [1]. The OE is a pseudostratified epithelium primarily consisting of sustentacular cells (SUSs), olfactory sensory neurons (OSNs), and neuroprogenitor and basal stem cells. SUS cell bodies in the OE are located superficially to form a protective sheet [2]. OSNs are bipolar neurons with an apical dendritic process extending through the SUS cell layer ending with anchored ciliary processes, carrying olfactory receptors and signaling molecules, which spread to cover the surface of the OE [3,4,5,6]. Therefore, both SUSs and OSNs are constantly exposed to potential environmental insults at the apical OE surface.

OSN axons, forming the olfactory nerve, connect the periphery to the olfactory bulb (OB) of the brain. Intranasal inoculation of several types of neurotrophic viruses, including influenza A virus, herpes simplex virus, hepatitis coronavirus, and vesicular stomatitis virus (VSV) was shown to infect OSNs and travel along the olfactory nerve to reach the brain [7,8,9,10,11,12]. Viral infection of the central nervous system (CNS) has dire consequences and often results in neuroinflammation, neuropathogenesis, and encephalitis [8,13,14]. The activation of astrocytes and microglia in the OB mount antiviral responses to prevent viral replication and viral spread into other regions of the brain [14,15,16].

While it is recognized that the OB can effectively curb viral replication and limit further spread into the CNS, it is not clear whether the OE, which directly interfaces with external environmental pathogens, mounts an effective innate immune response [15,17]. Neurons of the CNS have to carefully maintain a balance between inflammatory responses and suppression of cytolytic activity in order to protect neurons from viral-induced damage [18]. However, albeit being a neuronal tissue, the OE has a unique capacity for neurogenesis, rendering cell death a possible strategy for controlling viral infection [19]. The recruitment of phagocytic macrophages and apoptosis of OSNs were observed following pathogen challenge in the OE [20,21,22]. In addition, cytokine signaling was shown to participate in controlling viral infection in teleost fish OSNs as well as in mice [23,24].

The activation of an acute innate immune response is important in inhibiting viral replication and preventing viral spread. Though a few selected cytokines and signaling pathways have been characterized, systematic analyses for acute responses to viral infection in the OE are still lacking [21,23,25,26]. In this study, we characterized the transcription profile changes in the OE during the acute phase of VSV infection. Further characterization of cytokine and chemokine regulation revealed that both type I and type III interferon signaling are actively involved in the antiviral responses in the OE in a cell-type specific manner.

## 2. Materials and Methods

### 2.1. Animals

Adult C57BL/6J mice were obtained from Charles River Laboratories (Wilmington, MA, USA). *Ifnlr1* knockout mice (*Ifnlr1*^−/−^) were kindly provided by Dr. Herbert Virgin [26]. *Ifnar1* knockout mice (*Ifnar1*^−/−^) were acquired from MMRRC (B6.129S2-Ifnar1tm1Agt/Mmjax #32045-Jax). *Stat1* knockout mice (*Stat1*^−/−^) were obtained from Jackson Laboratory (B6.129S(Cg)-Stat1tm1Dlv/J #12606). All procedures were performed according to NIH guidelines and approved by the UC Davis Animal Care and Use Committee.

### 2.2. Vesicular Stomatitis Virus Nasal Inoculation

The original stock of VSV-12′GFP was kindly provided by Dr. Anthony van den Pol. This attenuated strain has two GFP genes inserted at the 3′ end of the viral genome, reducing VSV replication, but having no impact on the effectiveness of generating an immune response [27]. The VSV-12′GFP virus was replicated in BHK21 cells, concentrated, and purified into phosphate-buffered saline (PBS). Viral titers were determined using plaque assays [14,27]. For nasal inoculation, animals were deeply anesthetized using isoflurane. The VSV-12′GFP virus was instilled into the nasal cavity by placing drops of the virus in the nostril and allowing them to enter into the nasal cavity via inhalation. A total of 20 µL of 10^5^ pfu/μL virus was applied to each nostril for a total of 40 µL per animal. Similarly, phosphate-buffered saline (PBS) was administered to littermates as a vehicle control. We defined the acute phase of viral exposure as occurring within 48 h. Biological triplicates were collected at 3 h, 6 h, 12 h, 24 h, and 48 h post-viral inoculation (PI).

To administer the exogenous IFNs intranasally, 25 µL (12.5 µL for each nostril) of 40 ng/µL recombinant IFNβ1 (R&D, 8234-MB-010/CF) or IFNλ2 (R&D, 4635-ML-025/CF) was administered to mice one hour before VSV administration (20 µL of 10^6^ pfu/μL VSV at each nostril). Tissues were collected at 24 h PI to evaluate the exogenous IFNβ1 and IFNλ2 impact on VSV in the OE and OB.

### 2.3. RNA Sequencing

The OM was dissected and homogenized with TissueLyzer (Qiagen, Germantown, TN, USA) in TriZol. The total RNA was extracted and purified with a Zymogen RNA clean and concentration kit. The 3′-Taq-RNA-seq library was constructed, which generated a single initial library molecule per transcript. Biological triplicate samples were included for the PBS controls and VSV exposed at 3 h, 6 h, 9 h, 24 h, and 48 h PI. The libraries were sequenced on Illumina HiSeq 4000 with single-end 90 bp reads. A minimum of 6 million reads were obtained for each sample. After quality control, differential expression analyses were performed following the limma-voom Bioconductor pipeline. The comparisons between VSV and PBS at each time point are expressed as the log fold change (logFC) and the adj *p* values were determined using the Benjamini-Hochberg false discovery rate adjusted *p* value.

### 2.4. Western Blotting

The OM was dissected and flash-frozen in liquid nitrogen. RIPA buffer with a complete protease inhibitor was used to lyse the cells. Homogenization was performed using a TissueLyzer with 5 mm stainless steel beads (30 Hz, 1.5 min twice). Protein preparations were separated using a 12% SDS-PAGE gel and transferred to a nitrocellulose membrane. The antibodies used were: rabbit pSTAT1(Ser727) (0.2 μg/mL, Abcam ab109461 (Cambridge, MA, USA)), rabbit pSTAT2(Tyr689) (2 μg/mL, Millipore 07-224 (Bedford, MA, USA)), and Mouse α-tubulin (0.4 μg/mL, Active Motif 39528 (Carlsbad, CA, USA)).

### 2.5. Immunohistochemistry

Mice were perfusion-fixed with 4% paraformaldehyde followed by immersion post-fixation overnight at 4 °C. The OE and OB were cryoprotected in 30% sucrose, embedded in an optimal cutting temperature (OCT) compound, and sectioned at a thickness of 14 μm. The antibodies used were: chicken anti-OMP (custom, 1:1000), goat anti-GFP (Rockland (Baltimore, MD, USA), 2.2 µg/mL), rabbit anti-IFNAR1 (Sino Biological (Beijing, China), 1:20), goat anti-IFNLR1 (Novus Biologicals (Centennial, CO, USA), 10 μg/mL), and rabbit anti-pSTAT1 (Abcam, 3.0 µg/mL). The images were captured using an Olympus FV3000 confocal microscope.

### 2.6. Quantitative RT-PCR

The OM and OB were dissected and flash-frozen in liquid nitrogen post-viral infection. RNA extractions were performed with a Trizol and Zymogen RNA clean and concentrate kit. After reverse transcription with a poly deoxythymine (dT) primer, the cDNA was used for subsequent real-time PCR reactions. The real-time PCR experiments were conducted using SybrGreen chemistry and the applied biosystems StepOnePlus qPCR system. The differential gene expression was determined using the ΔΔCt method. The biological triplicates were included in all the RT-qPCR experiments. The primers used are listed in Table 1.

### 2.7. Apoptotic Cell Quantification

To quantify the apoptotic cells in the OE, the total volume of the OE was approximated by calculating the total number of slides, the number of sections per slide, and the size of each section for two sets of mice. The OE was sectioned at 14 μm with 4 sections mounted on each slide, resulting in a distance of 14 μm between the sections. An average of twenty-seven slides were used to section the entire OE. The total volume of the OE was then calculated as 14 μm × 4 sections per slide × 27 slides = 1512 μm total volume. The total volume of the OE was divided into three equal planes to represent the anterior, middle, and posterior of the OE. A slide was chosen from each plane of the OE, immunostained, and observed using a fluorescence microscope (Nikon Eclipse TE200 (Nikon Corporation, Tokyo, Japan)). For each slide, three randomly selected regions from two sections were chosen using DAPI, and the apoptotic cells which were immunopositive for activated caspase 3 were counted. The number of cells per mm was then extrapolated using Olympus FV31S confocal image analysis software.

## 3. Results

### 3.1. VSVs Infect Olfactory Sensory Neurons

To evaluate whether the VSV infects the OE, the VSV12′GFP virus was placed in the nostrils of mice and inhaled into the nasal cavity. Temporal and cell-type specific viral transduction were examined with viral GFP expression. No GFP expression was observed in the OE at 6 h post-viral instillation (6 h PI) (Figure 1A). Scattered GFP-positive cells were first observed at 12 h PI (Figure 1B). All GFP-positive cells observed at 12 h PI demonstrated the morphology of mature OSNs (Figure 1B). By 24 h PI, GFP-positive cells were widely distributed within the OE (Figure 1C). In addition to the OSNs, viral GFP expression was also observed in the sustentacular cells and basal cells. The OE tissue showed pathological changes, including a decrease in the OE thickness at 48 h PI with viral GFP expression still evident (Figure 1D). Viral GFP was observed in the olfactory nerve bundles within the lamina propria at 24 h PI (Figure 1C). The VSV-GFP-positive olfactory nerve continued into the olfactory nerve layer and glomerular layer of the OB (Figure 1E,F). Increased GFP expression levels were observed in the olfactory nerve layer and glomerular layers of the OB at 48 h PI. The GFP fluorescent signal was also co-localized with olfactory marker protein (OMP) immunoreactivity, labeling the olfactory axons and their terminal arbors within the glomeruli of the OB (Figure 1G,H). No VSV 12′GFP-positive OB neurons were observed at either 24 or 48 h PI [14].

To determine the presence of viral replication in the OM and OB, viral GFP transcript levels were measured using qRT-PCR and compared at 3 h, 6 h, 12 h, 24 h, and 48 h PI. The viral GFP transcripts were identified at 3 h PI, the earliest time point examined. The GFP transcript levels continued to increase in the OM from 6 h PI to 48 h PI (Figure 1I, relative Log10 fold change (LogFC): 3 h PI 2.34 ± 0.22; 6 h PI 2.75 ± 0.22; 12 h PI 4.67 ± 0.09; 24 h PI 6.87 ± 0.19; 48 h PI 7.4 ± 0.007, *p* < 0.05). In the OB, the viral gene transcripts were not detected at 12 h PI (Figure 1J, LogFC: 12 h PI –1.44 ± 0.58). Compared to the PBS control, VSV-GFP transcript levels were significantly increased at 24 h PI and 48 h PI (Figure 1J, LogFC: 24 h PI 4.26 ± 0.15; 48 h PI 5.4 ± 0.087, *p* < 0.05). During the acute phase of viral infection, while viral replication is active in OSNs, the rates of apoptosis were investigated. Activated caspase 3 immunohistochemistry was performed in the VSV-exposed OE (Figure 2). Increased numbers of apoptotic cells were observed compared to the wildtype control at 3 h, 6 h, 12 h, 24 h, and 48 h PI (Figure 2B, Cells/mm: wildtype 14.2 ± 1.7; 3 h PI 90.6 ± 2.7; 6 h PI 58.3 ± 4.7; 12 h PI 25.5 ± 1.2; 24 h PI 35.8 ± 10.8; 48 h PI 100 ± 2.4, *p* < 0.05). Interestingly, an increase in apoptosis occurred rapidly at 3 h PI and subsequently subsided before increasing again at 48 h PI, when the OE had undergone pathogenic changes.

### 3.2. Acute Changes of Transcription Profiles in the Olfactory Mucosa

During viral replication, apoptosis in the OE was relatively less robust between 3 to 48 h PI. To better understand the innate immune responses in the OM during the acute phase of VSV exposure, we conducted an RNA-seq analysis to systematically examine the gene expression changes within the first 48 h PI. We defined the period between viral exposure to 48 h PI as the acute phase. Biological triplicates of the OM were collected at 3 h, 6 h, 9 h, 24 h, and 48 h PI after VSV or PBS nasal inoculation. At 24 hr PI, 1655 transcripts showed differential expressions greater than 2.5 fold in the VSV samples compared to the PBS samples (Figure 3A, adj *p* < 0.05). The number of upregulated genes was higher than the number of genes downregulated (upregulated genes = 1230 genes; downregulated genes = 425 genes, adj *p* < 0.05). The genes showing robust differential expressions were cytokines and chemokines (Figure 3B,C). Significant changes were seen at 24 h PI; however, when evaluated using qRT-PCR, selected genes, such as *Il6*, showed upregulation at 3 h, which did not reach the significance level determined using adj *p* < 0.05 via RNA-seq (Figure 3E). Cytokine and chemokine transcriptional changes, specifically the upregulation of *Il6*, *Cxcl10*, and *Rela*, were further validated using qRT-PCR (Figure 3E–G). The gene ontology enrichment analysis was performed for differentially expressed genes for all of the time points examined. GO terms related to the antiviral responses were prominently recognized (Figure 3D).

### 3.3. Upregulation of Type I and III Interferon Transcript Levels

The gene ontology analysis identified a significant change in the cellular response to interferon-beta at 24 h PI (Figure 3D). In the mouse genome, there were fourteen interferon (*Ifn*) α isoforms, one *Ifnβ* isoform (type I), and two *Ifnλ* isoforms (type III). Neither type I nor type III *Ifn* expression was detected in the PBS-treated OM. Exposure to the VSV induced upregulation of *Ifn α2*, *α4*, *α16*, *β1*, *λ2*, and *λ3* at 24 h PI and 48 h PI in the OM compared to the PBS controls. The changes in the transcription of *Ifnα2*, *Ifnα4*, *Ifnβ1*, and *Ifnλ2/3* at 12 h, 24 h, and 48 h PI were validated using qRT-PCR (Figure 4A–D, *p* < 0.05). Type I and type III IFN receptors were expressed in the control OM. Compared to the PBS control OM, the type I *Ifn* receptor subunit (*Ifnar1*) and type III *Ifn* lambda receptor subunit (*Ifnlr1*) levels were not changed following VSV exposure at 24 h PI, as determined via RNA-seq analysis (Figure 4E, adj *p* < 0.05).

To determine the cell type-specific expression of *Ifn* receptors, immunohistochemistry was performed to detect the protein expression of IFNAR1 and IFNLR1. Mature OSNs were identified using olfactory marker protein (OMP) immunostaining. IFNAR1 expression was detected throughout the depth of the OE and appeared to be localized in the majority of the OM cell types, including OSNs (Figure 4F). IFNLR1 expression was less ubiquitous in the OE. Immunoreactivity of IFNLR1 in the sustentacular cell body layer at the apical surface of the OE was not evident. However, IFNLR1 expression was detected in OMP-positive OSNs throughout the OE (Figure 4G). IFNLR1 expression was also observed in the olfactory nerve bundles, labeled via OMP immunoreactivity, in the lamina propria. Therefore, IFNLR1 was specifically expressed in mature OSNs and not at the detection level in other cell types of the OE.

### 3.4. Activation of Interferon Signaling in the Olfactory Epithelium

The activation of type I and type III *Ifn* receptors results in the phosphorylation of *Stat1* (pSTAT1) and *Stat2* (pSTAT2) and the subsequent regulation of *Ifn-stimulated* genes (ISGs), which perform antiviral functions [28,29]. To investigate whether exposure to the VSV activates *Ifn* signaling in the OM, we first examined pSTAT1 and pSTAT2 levels using Western blotting. pSTAT1 and pSTAT2 were not detected in the PBS control OM, while they were clearly present in the VSV-exposed OM at 24 h PI (Figure 5A,B). The pSTAT1 and pSTAT2 levels were quantified and normalized against an α-tubulin loading control. Comparisons of the VSV to PBS samples in biological-triplicate-OMs showed significant changes in pSTAT1 and pSTAT2 at 24 h PI (pSTAT1 PBS vs. VSV: 0.063 ± 0.019 vs. 0.30 ± 0.049; pSTAT2 PBS vs. VSV: 0.21 ± 0.009 vs. 0.46 ± 0.06, Student’s *t*-test, *p* < 0.005).

To investigate cell type-specific STAT activation, immunocytochemistry was performed to detect the pSTAT1 expression in the OE. pSTAT1 was detected in the majority of cell types, including OSNs, at low levels in the PBS control. At 24 h PI VSV, pSTAT1 appeared in the nuclei of the majority of cell types and robustly in sustentacular cells (Figure 5C).

The expression levels of ISGs were evaluated in the VSV-exposed OM and compared with the PBS controls using qRT-PCR (Figure 5D–F). The upregulation of *Oas1*, *Ifit2*, and *Ifit3* transcript levels were first detected at 12 h PI (Fold: 12 h PI: *Oas1* 2.14 ± 0.075, *Ifit2* 1.83 ± 0.06, *Ifit3* 6.5 ± 0.05; 24 h PI: *Oas1* 29.8 ± 0.13, *Ifit2* 16.9 ± 0.11, *Ifit3* 93.1 ± 0.11; 48 h PI: *Oas1* 28.49 ± 0.13, *Ifit2* 23.95 ± 0.14, Ifit3 86.6 ± 0.13 *p* < 0.05). Increased levels of upregulation of ISGs were observed at 24 h and 48 h PI compared to that of 12 h PI.

Interferon signaling was required for suppressing VSV replication in the olfactory mucosa. To determine whether *Ifn* signaling is required for performing antiviral functions in the OM, we examined VSV viral load in *Ifnar1* and *Ifnlr1* knockout mice and compared it to *Ifnar1* and *Ifnlr1* wildtype OM. The relative expression levels of the three viral genes VSV-GFP, VSV-M, and VSV-N were measured at 24 h PI. In the *Ifnar1* knockout OM, all three examined viral genes were slightly increased relative to the wildtype, but the change was not significant. (Figure 6A, relative fold *Ifnar*^−/−^: VSV-GFP: 1.43 ± 0.042; VSV-M: 1.48 ± 0.03; VSV-N: 1.54 ± 0.052, *p* > 0.05).

Similarly, in the *Ifnlr* knockout OM, slight non-significant increases were observed in VSV viral gene expression compared to the wildtype (Figure 6B, relative fold *Ifnlr*^−/−^: VSV-GFP: 1.57 ± 0.09 VSV-M: 1.27 ± 0.12; VSV-N: 1.4 ± 0.07, *p* > 0.05). However, in the OM from *Ifnar1*^−/−^/*Ifnlr1*^−/−^ double knockout mice, significant increases in viral gene expression were detected with qRT-PCR (Figure 6C, relative fold to PBS: VSV-GFP: 2.4 ± 0.51 VSV-M: 3.5 ± 0.47; VSV-N: 3.43 ± 0.41 *p* < 0.05). Furthermore, knocking out *Stat1*, which disrupts both type I and type III *Ifn* signaling, also resulted in a significant increase in viral gene expression (Figure 6D, relative fold: VSV-GFP: 2.87; VSV-M: 5.71; VSV-N: 3.97, *p* < 0.05). These results indicate that both type I and type III *Ifn* signaling are required for viral replication suppression in the OM.

Furthermore, we investigated whether IFNs are sufficient in reducing viral load in the OM. Exogenous IFNβ1, IFNλ2, or PBS was administered via nasal instillation to wildtype mice 1 h before VSV exposure. Viral transcript levels were examined at 24 h post-viral inoculation and compared between the PBS and IFN groups in biological triplicates. It was observed that the administration of IFNβ1 decreased the relative number of viral transcripts of all three viral genes examined in the OM (Figure 7A, relative fold to PBS: VSV-GFP: 0.18 ± 0.26 VSV-M: 0.21 ± 0.27; VSV-N: 0.19 ± 0.26). Consistently, the viral transcript levels decreased in the OB at 24 h PI as well as when primed with IFNβ1 (Figure 7B, relative fold to PBS: VSV-GFP: 0.26 ± 0.27 VSV-M: 0.27 ± 0.27; VSV-N: 0.26 ± 0.27). When comparing priming with exogenous IFNλ2 to PBS before VSV exposure, decreases in viral transcripts were observed at 24 h PI for all three viral genes in the OM (Figure 7C, relative fold to PBS: VSV-GFP: 0.17 ± 0.07; VSV-M: 0.12 ± 0.06; VSV-N: 0.17 ± 0.08). Viral transcript levels also decreased consistently at 24 h PI in the OB following IFNλ2 priming of the OE (Figure 7D, relative fold to PBS: VSV-GFP: 0.2 ± 0.44 VSV-M: 0.29 ± 0.28; VSV-N: 0.65 ± 0.25). Though consistent viral load reductions were observed in both cases, the changes did not reach statistical significance in both the OE and OB. Nonetheless, these results indicate that type I and type III IFNs are sufficient to suppress viral replication in the OE. VSV-GFP expression was examined in the OE sections. Consistent with qRT-PCR, VSV-GFP was diminished in IFNβ1-primed OE. GFP-positive cells were scarcely scattered in the basal cell layer of the OE in IFNλ2-treated animals (Figure 7E–G). This data may reflect the expression differences between type I and III receptors.

## 4. Discussion

Despite its vulnerability to pathogen exposure, the mammalian olfactory mucosa’s innate immune responses to viral infection have not been characterized systematically. In this study, we report that VSV infects the majority of cell types in the OE. Acute transcriptional responses were observed as early as 3 h post-VSV nasal infection. Following VSV exposure, both type I and type III IFNs were upregulated. The activation of their signaling pathways was detected within the OM. The genetic analyses indicated that IFN signaling is required to control VSV viral load in the OM.

The glycoprotein G of the VSV is widely used to pseudotype other viruses for gene transfer. The receptor for viral entry of the VSV, the low-density lipoprotein receptor (*Ldlr*), is expressed in the olfactory mucosa. It is expected that VSVs are able to infect cells in the OE. Nasal instillation of the VSV results in viral entry into the OB, likely through the olfactory nerve [14]. Consistent with previous studies, VSV infection and viral replication in OSNs were observed and were widespread at 24 h PI [11]. The attenuated VSV strain used in this study, VSV12′GFP, replicated slower than the wildtype Indiana strain. Nonetheless, we detected GFP expression as early as 12 h PI in the OSNs, suggesting that the virus could efficiently enter OSNs and take over the cellular machinery of the OM to replicate. Viral GFP expression in sustentacular cells as well as basal cells was not detected until 24 h PI. GFP could be detected in olfactory nerve bundles within the OM as well as in the olfactory nerve layer and in the terminals of the olfactory nerve within the glomeruli of the OB. These observations are consistent with studies suggesting that instead of packaging in the cell body, VSVs transport their viral proteins to the nerve terminals and assemble into virions locally for shedding [30]. For the time points analyzed up to 48 h PI, we did not detect any viral GFP in OB cells. Further experiments are needed to determine whether VSV12′GFP spread is effectively blocked in the OB [14].

Limiting viral spread in the OE is particularly critical due to its direct connection with the CNS. It has been established that viral exposure triggers acute innate immune responses, including apoptosis, proinflammatory cytokines, and chemokines in many tissue types [31,32]. OSNs are a unique population of neurons, which possess a limited lifespan and are replaced by basal stem cells in the OE [19,33]. When OSNs reach the end of their life or are damaged by environmental insults, they die through apoptosis [34,35]. Increased OSN apoptosis and replacement are speculated to be a protective mechanism for the CNS. Viral infection-induced apoptosis is a way for a virus to evade the host’s innate immune mechanism for its survival [36]. The VSV is known to induce the rapid apoptosis of the host’s cells as early as 9.5 h post-infection in HeLa cells in vitro and at 24 h in vivo [37,38]. In our study, we observed increased apoptosis at 3 h post-infection. This is the earliest onset of viral-induced apoptosis observed in the OSNs prior to the onset of the antiviral responses in the olfactory mucosa. The second peak of apoptosis at 48 h possibly reflects continuous pathogenesis as a result of VSV survival.

To systematically evaluate the dynamics of transcriptional responses, we performed an RNA-seq analysis of the whole OM harvested at different time points post-viral exposure. It is evident that the majority of cytokines and chemokines are either not expressed or expressed at very low levels at resting state. Compared to vehicle controls with three biological replicates, we did not detect significant transcriptional changes before 24 h PI. When selected cytokines were examined using qRT-PCR, we did detect significant increases in *Il6* expression at 3 h PI. *Il6* transcripts were upregulated at 3 h PI in the RNA-seq dataset but did not reach statistical significance (LogFC = 1.8 VSV/PBS, *adj p* = 0.76). It is possible that transcriptional changes occurred in a small subset of cells and, therefore, the differential expression was not detectable at significant levels when analyzing the global changes. Further studies are needed to dissect the cell type-specific contributions to these gene expression changes.

IFNs are antiviral cytokines that are critical in cellular defense mechanisms against viral infections. *Ifn* signaling is particularly important in neurons as they are postmitotic and not typically replaced following damage. OSNs are a special type of neuron, which can be replaced throughout an animal’s lifetime. It remains unknown whether OSNs utilized the same strategy to suppress viral replication and protect from cell death [8,39]. *Ifng* signaling is critical for neuronal protection in the CNS [39]. In this study, *Ifng* expression was not detected in either the control or VSV-stimulated OE. Among the 13 type I *Ifn* isoforms, *Ifnβ1* is prominently upregulated as well as type III Ifns (*Ifnλ2* and *Ifnλ3*), suggesting both type I and type III *Ifns* are involved in VSV-induced responses. Consistent with many tissue types, the type I *Ifn* receptor, *Ifnar1*, is ubiquitously expressed in the OE. On the contrary, the type III *Ifns* receptor, *Ifnlr1*, is specifically expressed in mature OSNs. The presence of both type I and type III *Ifns* receptors suggests that OSNs are equipped with antiviral protection from both *Ifn* types. With the upregulation of *Ifnβ1* and *Ifnλ2/3*, we expected that the loss of function of either IFNAR1 or IFNLR1 would result in reduced antiviral activity. However, neither the IFNAR1 nor IFNLR1 knockout resulted in an increased viral copy number in the OE when compared to the vehicle control. Only in *Ifnar1*^−/−^/*Ifnlr1*^−/−^ double knockout mice, did VSV load increase in the OE following VSV challenge. This suggests that *Ifn* signaling indeed inhibits VSV replication in the OE. The lack of evident antiviral effect in the single IFNAR1 or IFNLR1 receptor knockout mice could be the result of *Ifn* signaling compensation between the two different types. It is also possible that the type I and type III *Ifn* levels are relatively low and, therefore, are not able to exert consistent detectable differences by themselves. Both type I and type III IFN receptor activation results in the phosphorylation of STAT1, which subsequently activates transcription [40,41]. Consistent with the receptor double knockout results, we also observed decreased viral replication inhibition in the *Stat1*^−/−^ OE. These observations indicate that an effective antiviral response utilizes both type I and type III IFN signaling pathways.

IFNλ is produced in many tissues in response to viral infection. Though IFNλ is believed to regulate the same set of genes as type I IFNs, IFNLR1 expression is limited to certain cell types and, therefore, functions in a tissue-specific manner [42]. The expression of type I and type III receptors is not upregulated after viral exposure. Activation of IFN signaling is controlled by the upregulation of IFN isoforms. We noticed that pSTAT1 was localized in the OSNs in the control OE. This may indicate that OSNs maintain a heightened antiviral state. With VSV exposure, however, pSTAT1 was more prominent in the nuclei of SUS cells, suggesting rapid protective responses by SUS cells. The impact of IFN antiviral function is evident when exogenous IFNβ1 or IFNλ2 were administered prior to VSV infection. The robust effect of exogenous IFNs on suppressing viral replication in the OE indicated that the interferon signaling pathway was in place and could be utilized to protect the OE. It is still unclear which cell types in the OE produce IFNs upon viral insult and whether IFN signaling protects OSNs from undergoing apoptosis.

## Figures and Tables

**Figure 1 biomolecules-13-01762-f001:**
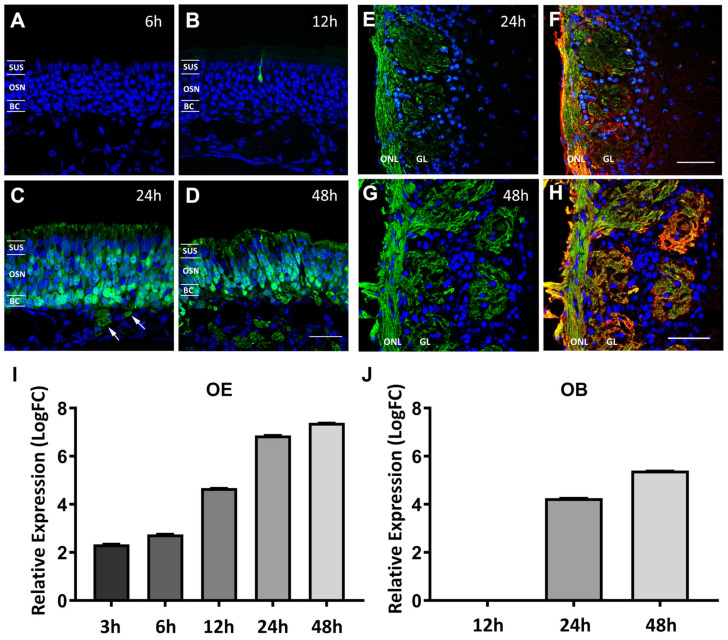
VSVs infect olfactory sensory neurons. Viral GFP expression in the OE at 6 h (**A**), 12 h (**B**), 24 h (**C**), and 48 h (**D**) after VSV12′GFP nasal instillation. The locations of the sustentacular cell nuclei layer (SUS), olfactory sensory neurons (OSNs), and basal cells (BCs) are marked (**A**–**D**). Olfactory axons positive for viral GFP were observed in the lamina propria at 24 h and 48 h PI ((**C**,**D**), arrows), as well as in the olfactory nerve layer (ONL) and glomerular layer (GL) of the OB (**E**–**H**). OMP immunostaining (red, in (**F**,**H**)) outlines the distribution of the olfactory axons. Relative expression of viral GFP transcripts in the OE (**I**) and OB (**J**) at different time points after viral exposure, determined using qRT-PCR, demonstrates a viral load increase in the tissue. Scale bar = 30 μm in (**D**) and 50 μm in (**F**) and (**H**).

**Figure 2 biomolecules-13-01762-f002:**
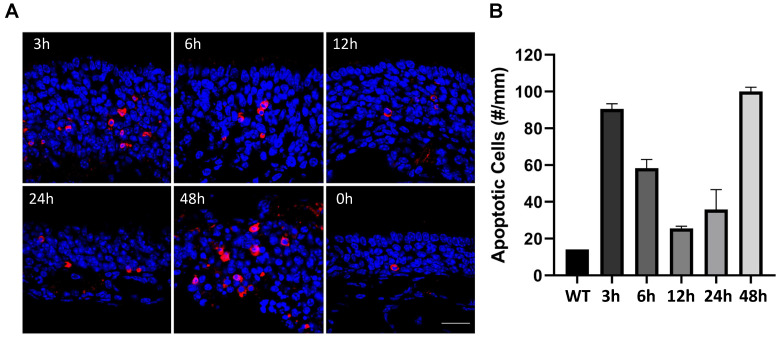
Viral infection induces apoptosis in the olfactory epithelium. Activated caspase-3 cells showing apoptosis in the OE at 0 h, 3 h, 6 h, 12 h, 24 h, and 48 h post-VSV infection (**A**). Apoptotic cell count per mm of OE tissue at 3 h, 6 h, 12 h, 24 h, and 48 h PI (**B**). Scale bar = 20 μm.

**Figure 3 biomolecules-13-01762-f003:**
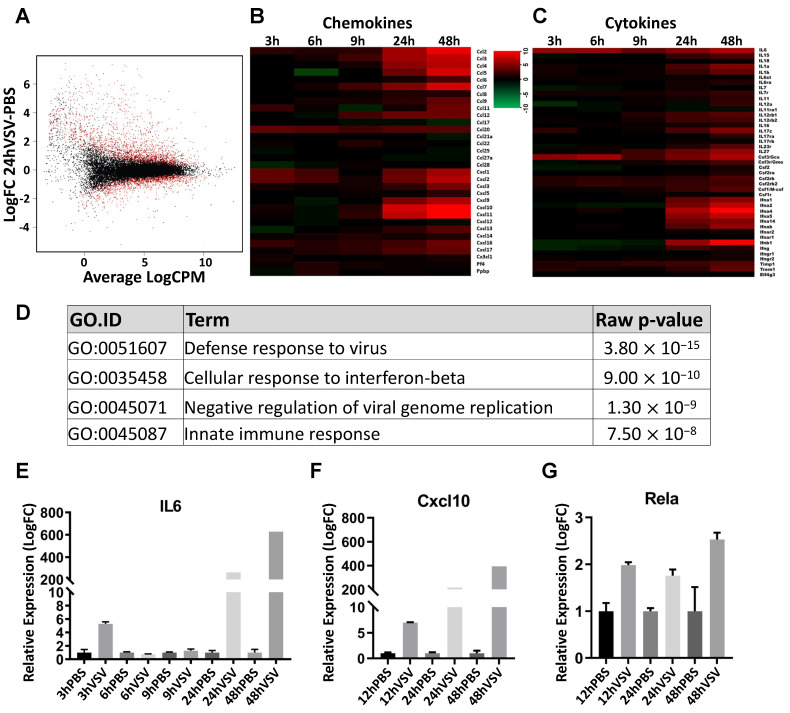
VSV exposure triggers acute changes in transcription profiles in the olfactory mucosa. Scatter plot comparing gene expression differences between the VSV and PBS-exposed olfactory mucosa at 24 h PI via RNA-seq (**A**). The red dots represent the genes with adj *p* < 0.05. Heatmap of chemokines and cytokine levels at 3 h, 6 h, 9 h, 24 h, and 48 h PI (**B**,**C**). Significant associations of gene ontology (GO), particularly antiviral responses, among differentially expressed transcripts between VSV and PBS control olfactory mucosa at 24 h PI (**D**). Upregulation of Il6, Cxcl10, and Rela at 3 h, 6 h, 12 h, 24 h, and 48 h PI via qRT-PCR (**E**–**G**).

**Figure 4 biomolecules-13-01762-f004:**
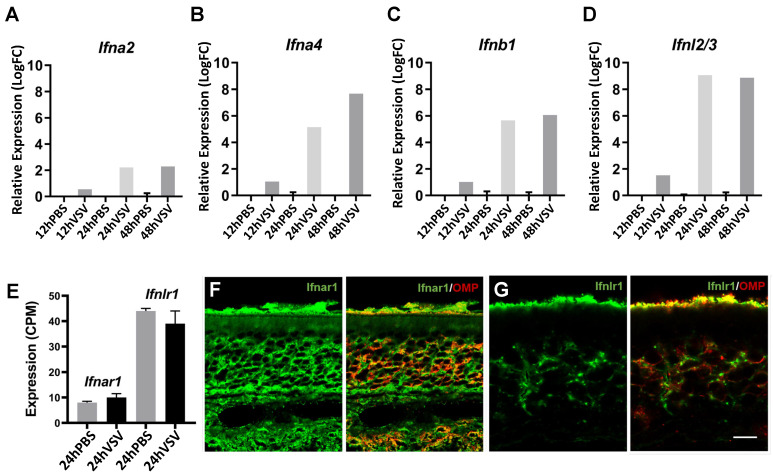
Upregulation of type I and III interferon transcript levels. Relative expression of Ifna2, Ifna4, Ifnb2, and Ifnl2/3 in the olfactory mucosa using qRT-PCR (**A**–**D**). Relative expression of Ifnar1 and Ifnlr1 in the olfactory mucosa at 24 h PI (**E**). Biological triplicates were included (**A**–**E**). Immunostaining of IFNAR1 (green in (**F**)) and IFNLR1 (green in (**G**)) with OMP (red in (**F**,**G**)) in the OE. Bar = 15 μm.

**Figure 5 biomolecules-13-01762-f005:**
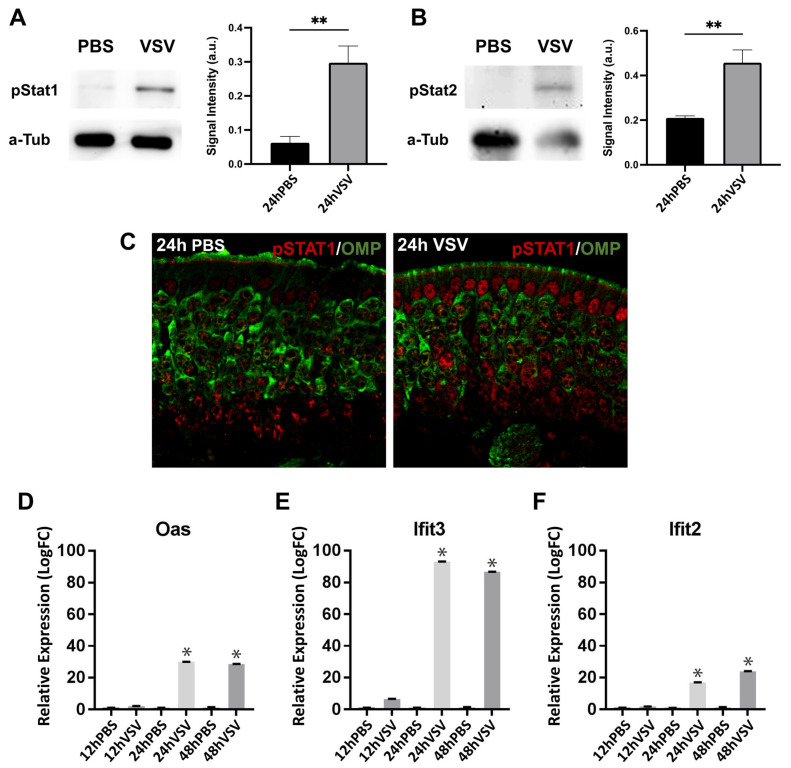
Activation of interferon signaling in the olfactory epithelium upon VSV exposure. pSTAT1 and pSTAT2 levels in the olfactory mucosa were examined using Western blotting at 24 h PI, with signals quantified against an a-tubulin loading control. Biological triplicates were included for quantification (**A**,**B**). Immunohistochemistry of pSTAT1 in the 24 h PI OE (**C**). Relative expression of interferon-stimulated genes, Oas, Ifit2, and Ifit3 were determined using qRT-PCR (**D**–**F**). Biological triplicates were included. Student’s *t*-test, ** *p* < 0.005, * *p* < 0.01. Original western blot images of Figure 5 can be found in the Appendix A.

**Figure 6 biomolecules-13-01762-f006:**
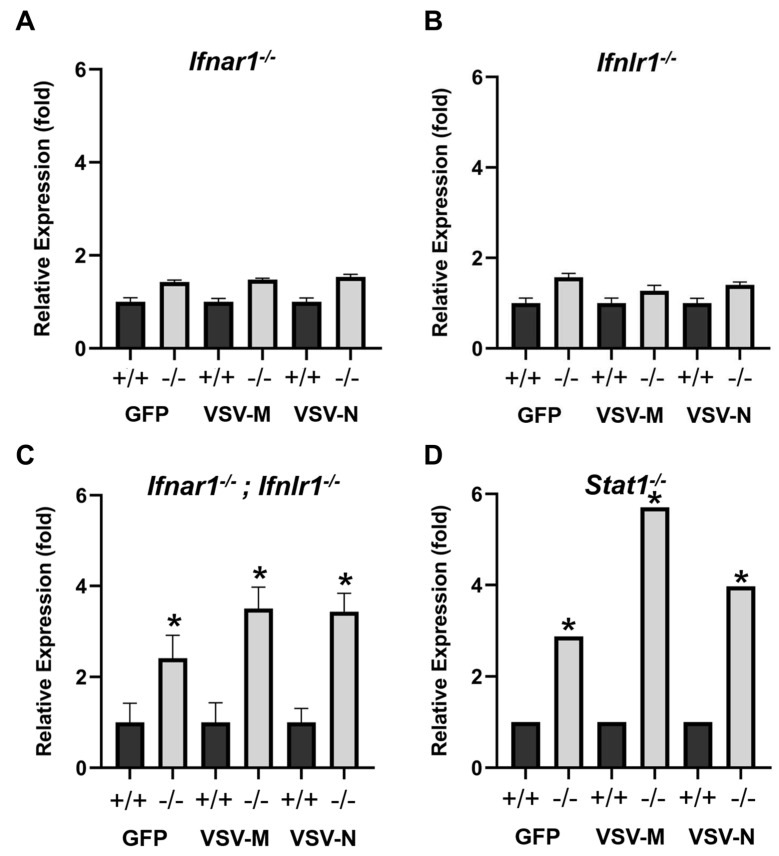
Interferon signaling is required for suppressing VSV replication in the olfactory mucosa. The expression levels of the viral genes, VSV-GFP, VSV-M, and VSV-N, in olfactory mucosae at 24 h PI were measured in *Ifnar1*^−/−^ (**A**), *Ifnlr1*^−/−^ (**B**), *Ifnar1*^−/−^; *Ifnlr1*^−/−^ (**C**), and *Stat1*^−/−^ (**D**) and compared to wildtype littermates. Student *t*-test, * *p* < 0.05.

**Figure 7 biomolecules-13-01762-f007:**
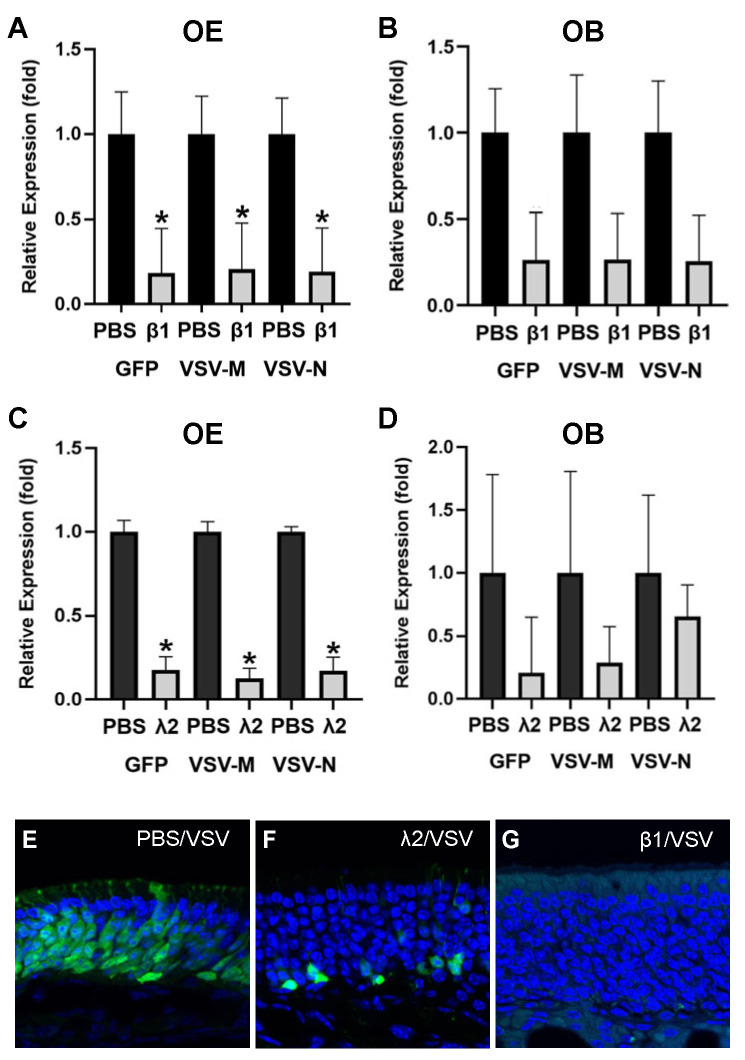
Type I and III interferons are sufficient to suppress viral load in the olfactory mucosa. Antiviral functions of type I and III interferons were examined by providing exogenous recombinant mouse interferon β1 (**A**,**B**,**G**) and interferon λ2 (**C**,**D**,**F**). PBS was instilled as the control. Interferon or PBS was provided an hour before VSV administration. VSV-GFP, M, and N gene expression levels were measured 24 h PI in the OE (**A**,**C**) and OB (**B**,**D**). VSV-GFP protein reduction was observed in the OE 24 h PI for exogenous interferon λ2 (**F**) and interferon β1 (**G**) compared to PBS control (**E**). Student’s *t*-test, * indicated *p* < 0.05.

**Table 1 biomolecules-13-01762-t001:** List of qPCR primers.

Primer Name	Sequence
VSV-GFP	F: GAGCGCACCATCTTCTTCAAG R: TGTCGCCCTCGAACTTCAC
VSV-M	F: TCGGTCTGAAGGGGAAAGGT R: AGGTGTCCATCTCGTCAACTC
VSV-N	F: GATAGTACCGGAGGATTGACGACTA R: TCAAACCATCCGAGCCATTC
Ifna2	F: TGCTTTCCTCGTGATGCTGA R: TCATCTGTGCCAGGACCTTC
Ifna4	F: GCCTTGACAGTCCTGGAAGA R: TTGAGCTGCTGATGGAGGTC
Ifnb1	F: CAGCTCCAAGAAAGGACGAAC R: GGCAGTGTAACTCTTCTGCAT
Ifnl2/3	F: AGCTGCAGGCCTTCAAAAAG R: TGGGAGTGAATGTGGCTCAG
Oas	F: GATGTCAAATCAGCCGTCAA R: AGTGTGGTGCCTTTGCCTGA
Ifit2	F: AGTACAACGAGTAAGGAGTCACT R: AGGCCAGTATGTTGCACATGG
Ifit3	F: GGGAAACTACGCCTGGATCTACT R: CATGCTGTAAGGATTCGCAAAC
Il6	F: ATGATGGATGCTACCAAACTGGA R: CTGAAGGACTCTGGCTTTGTCT
Cxcl10	F: ATCATCCCTGCGAGCCTATCCT R: GACCTTTTTTGGCTAAACGCTTTC
Rela	F: CTGCCGAGTAAACCGGAACT R: GCCTGGTCCCGTGAAATACA
Gapdh	F: TGCACCACCAACTGCTTAG R: GGATGCAGGGATGATGTTC

## Data Availability

RNA-seq data presented in this study are available at Gene Expression Omnibus # GSE154235.

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
