# Peer review of "Antiviral Functions of Type I and Type III Interferons in the Olfactory Epithelium"

_biomolecules, 2023, doi:10.3390/biom13121762_

Round 1
Reviewer 1 Report
Comments and Suggestions for Authors
Manuscript review: biomolecules-2722114 “Antiviral functions of type I and type III interferons in the olfactory epithelium.” Corresponding author: Qizhi Gong
The authors investigate immune response to viral infection in the olfactory epithelium. Based on differential gene expression, they identify a role for type I and III interferons in suppression of viral replication.
Comments:
1. Figure 1: Please stain for or otherwise label sus cells, basal cells, etc.
2. The authors briefly touch on the biphasic kinetics of cellular apoptosis in Figure 2B. Could the authors please expand upon why that may be?
3. Figure 4: The authors show more abundant expression of IFNLR1 compared to IFNAR1 in the OM (Fig 4E), however 4F seems to show the opposite. Why is that?
4. Figure 6: This experiment should include results from a WT control mouse.
5. Please detail the types of statistical analyses used in each figure. I am surprised that Fig 7D has significance given the size of the error bars in both control and lambda2 samples.
Minor comments:
1. Please list concentrations of antibodies used rather than dilutions. Please also report antibody concentrations for Western blots.
2. I might have missed it but I could not find a definition for the ‘P’ in the acronym OMP. Please clarify.

Author Response
We appreciate all the critiques and comments. Revisions are made to address all the critiques. The manuscript is improved as a result.

Reviewer 2 Report
Comments and Suggestions for Authors
Vesicular stomatitis virus (VSV), a known neurotropic virus that can travel to and propagate in brain tissues, establishes productive infections in mice that are inoculated via a nasal/olfactory route. In this manuscript, the authors provide compelling evidence that this mode of infection occurs in the olfactory neuroepithelium, specifically in olfactory sensory neurons (OSNs) which are central players involved in limitation of the infection. How the virus alters the microenvironment within infected mice remained unknown along with the antiviral responses that are induced upon infection. To this end the authors employed a murine model of infection using a VSV strain that expresses eGFP thus allowing the tracking of viral infection. Using this model, the authors provide compelling evidence that infection initiates in olfactory epithelium (OE) at first then eventually resulting in productive infection within the olfactory bulb. The OE infection induced an apoptotic response. Few neuronal cells are amenable to apoptotic events as they are, for the most part, non-replicative, except for the OSNs which can self-renew. Thus, they are capable of undergoing apoptosis with a replacement resulting in little damage to the host.
To further investigate the impact of viral infection, the authors utilized an unbiased RNA-seq transcriptome profiling approach to identify transcriptional changes in the OE and found a significant increase in chemokine and cytokine responses upon infection in time frames consistent with viral spread in the tissues, which were validated by RT-qPCR. A significant increase in interferon responsive genes were identified which led to investigations of type I and type III interferon responses being further investigated. The authors utilized knockout mice to show that inhibition of the receptors for both INF I and INF III pathways (but not those individually) were sufficient to limit viral spread as was knockout of the central signaling factor for both pathways STAT1. Finally, the authors show that exogenous treatment with interferon Type I or Type III was sufficient to limit VSV-eGFP spread.
While the findings are not overly surprising, the study is well performed, the conclusions are supported by the offered results and the experiments are sufficiently controlled. As such, only minor comments are offered to increase the readability of the study.
Minor Comments:
1) While stated in the text for some of the experiments (but not all) there would be value to including the number of biological replicates used for each of the experiments shown. This is particularly important when looking at representative images such as figure 1, 2, 4, 5 and 7.
2) In Fig 3D, it is not apparent if the GO.ID terms shown were the top hits or selected term hits.
3) The densometry in Fig 5a lacks error bars (or maybe they are so tight they are hard to see) but in three biological replicates I would be inclined to believe there should be some variation.
4) Results of the RNA-seq needs to be deposited in an accessible repository.
Author Response

(The authors gave the same response as above.)
